# Bacteriostatic Behavior of PLA-BaTiO_3_ Composite Fibers Synthesized by Centrifugal Spinning and Subjected to Aging Test

**DOI:** 10.3390/molecules26102918

**Published:** 2021-05-14

**Authors:** Francesco Boschetto, Hoan Ngoc Doan, Phu Phong Vo, Matteo Zanocco, Kenta Yamamoto, Wenliang Zhu, Tetsuya Adachi, Kenji Kinashi, Elia Marin, Giuseppe Pezzotti

**Affiliations:** 1Ceramic Physics Laboratory, Kyoto Institute of Technology, Sakyo-ku, Matsugasaki, Kyoto 606-8585, Japan; matteo.zanocco@gmail.com (M.Z.); wenlzhu@hotmail.com (W.Z.); elia-marin@kit.ac.jp (E.M.); pezzotti@kit.ac.jp (G.P.); 2Department of Dental Medicine, Graduate School of Medical Science, Kyoto Prefectural University of Medicine, Kamigyo-ku, Kyoto 602-8566, Japan; fiori30@koto.kpu-m.ac.jp (K.Y.); t-adachi@koto.kpu-m.ac.jp (T.A.); 3Functional Polymer Design Laboratory, Kyoto Institute of Technology, Sakyo-ku, Matsugasaki, Kyoto 606-8585, Japan; ngochoandoan@gmail.com (H.N.D.); vpphu94@gmail.com (P.P.V.); kinashi@kit.ac.jp (K.K.); 4Department of Immunology, Kyoto Prefectural University of Medicine, Kamigyo-ku, Kyoto 602-8566, Japan; 5Department of Orthopedic Surgery, Tokyo Medical University, 6-7-1 Nishi-Shinjuku, Shinjuku-ku, Tokyo 160-0023, Japan; 6The Center for Advanced Medical Engineering and Informatics, Osaka University, 2-2 Yamadaoka, Suita, Osaka 565-0854, Japan

**Keywords:** bacteriostatic behavior, polylactic acid, composite fibers, barium titanate, neutralization

## Abstract

The present work investigated the effect of Polylactic acid (PLA) fibers produced by centrifugal spinning with incorporated BaTiO_3_ particles to improve their bacteriostatic behavior. The PLA matrix and three composites, presenting three different amounts of fillers, were subjected to UV/O_3_ treatment monitoring the possible modifications that occurred over time. The morphological and physical properties of the surfaces were characterized by different microscopic techniques, contact angle, and surface potential measurements. Subsequently, the samples were tested in vitro with human dermal fibroblasts (HDF) to verify the cytotoxicity of the substrates. No significant differences between the PLA matrix and composites emerged; the high hydrophobicity of the fibers, derived by the polymer structure, represented an obstacle limiting the fibroblast attachment. Samples underwent bacterial exposure (*Staphylococcus epidermidis*) for 12 and 24 h. Increasing the concentration of BT, the number of living bacteria and their distribution decreased in comparison with the PLA matrix suggesting an effect of the inorganic filler, which generates a neutralization effect leading to reactive oxygen species (ROS) generation and subsequently to bacterial damages. These results suggest that the barium titanate (BT) fillers clearly improve the antibacterial properties of PLA fibers after aging tests made before bacterial exposure, representing a potential candidate in the creation of composites for medical applications.

## 1. Introduction

Polylactic acid (PLA) is a biodegradable polymer widely used in the development of tissue engineering, biodegradable implants, and the control of drug release rate [1,2,3,4]. Furthermore, it possesses other excellent characteristics such as good mechanical properties, thermal plasticity, processability, and high water resistance [5,6,7,8]. However, it also presents disadvantages, such as low resistance to aging and low antibacterial properties. This latter, in particular, represents still the major problem and can influence its applicability in the medical field. One way of improvement focused on creating composites incorporating inorganic particles that possess an effect against bacteria through different pathways. For example, Ag nanoparticles have the ability to kill a great variety of Gram-positive and Gram-negative bacteria [9]; also, titanium dioxide (TiO_2_) particles have been widely investigated due to their antimicrobial property caused by photocatalysis. When this latter occurred, under light, it generates enough energy for the production of reactive oxygen species (ROS), which possess great potential in damaging microbial cells [10,11]. Another inorganic filler used as an alternative antimicrobial agent is the barium titanate (BT). Shah et al. studied the effect of BT, discovering its biofilm inhibition activity against Gram-negative and Gram-positive bacteria [12].

The development of scaffolds using some inorganic particles has attracted a lot of attention, and several techniques have been used to obtain these composites depending on the applications [13,14]. One of these technologies is electrospinning that generates micro/nanofibers on a large scale and at a low cost starting from polymeric solutions [15,16,17,18,19,20]. Although this technique is highly versatile, it presents some limitations as the conductivity of the solution, a high-voltage electric field necessary, and a low production rate. To avoid these conditions, centrifugal spinning, an alternative technique, has been developed. It provides efficient fiber production with a low cost at higher rates and insensitivity to the dielectric constant of materials [21,22,23,24,25,26,27,28]. This method briefly consists of injecting the material into a rotating spinneret with one or multiple nozzles, from which the material is expelled when the centrifugal force is larger than the surface tension of the material. Fibers are produced through stretching by inertial forces as the solvent is evaporated, based on the cotton-candy production principle [29,30]. Properties such as nozzle geometry, rotation speed, and polymer solution can control the polymer fiber porosity and diameter. Since this technique does not use a high-voltage electric field, it alleviates the related safety concern; furthermore, the high rotational speed allows fast and scalable fiber fabrication, which can dramatically improve the production rate by two to three orders of magnitude. The use of PLA to produce composites with antimicrobial properties by centrifugal spinning has been already described in the literature [31]; however, the bacteriostatic behavior of composites fibers has never been studied after the aging test in order to verify if the capacity to resist to bacterial adhesion could be durable even after physical modifications over time.

In our previous work, the addition of BT provided a more resistant structure against UV/O_3_ used for aging tests due to an improvement connected to a variety of crystalline phase and chemical structure which conferred stability during the physical treatment [32]. The same types of samples have been an object of study in this work, focusing attention on the antibacterial properties of the composites and analyzing their durability over time in correlation with the morphological and physical changes at the surface level brought by the treatment. The samples have been characterized by different techniques and subsequently treated with fibroblasts and one bacterial Gram-positive strain in order to monitor the cytotoxicity and the bacteriostatic behavior over time provided by the BT.

## 2. Results

### 2.1. Characterization

The laser micrographs shown in Figure 1, obtained at 10× of magnification, revealed how the fibers are distributed on all four samples before the aging treatment. The PLA matrix and PLA 5% BT surfaces presented a random orientation of the fibers, while the other two samples presented a more oriented direction. Furthermore, PLA 10% BT and PLA 15% BT showed an increase in fiber diameter while, in the other two classes, the fibers exhibited different sizes. Few clusters related to the drying of polymers droplets on the tip of the needle during the electrospinning process are visible, except for the PLA.

Studying the profilometry of images collected at 150× of magnification, the average roughness value, Ra, and mean roughness depth, Rz of the four tested surfaces were calculated and their differences were statistically validated. The values are reported in Figure 2. A decrease of Ra is visible when implementing the amount of barium titanate. No-aging treated PLA 15% BT showed lower values than the other samples, especially PLA, which possessed the highest one. An important feature that emerged is that the aging treatment did not influence the roughness of PLA 10% BT and 15% BT; in fact, no statistical variations have been reported over time during the UV/O_3_ exposure for these two classes. Instead, in PLA and PLA 5% BT, a reduction of Ra has been detected after 10 min of UV/O_3_ treatment. Rz values did not show the same trend. Comparing the four untreated substrates, no statistical differences were observed. PLA 10% BT and PLA 15% BT also did not show any variations connected with the aging test. PLA 5% BT treated for 5 min exhibited the highest Rz, while at 10 min, the value was in the same range as the ones detected for PLA 10% BT and PLA 15% BT. Considering PLA without any fillers, the analysis showed a clear decrease over time of aging treatment, especially at 10 min.

Figure 3 shows representative images of water contact angles as measured on the four samples. The graph displays how the PLA substrates without fillers were the least hydrophobic and did not present any statistical difference despite the treatment. Furthermore, each group of composites did not show variation related to the UV/O_3_ action.

No differences were observed between the PLA matrix, PLA 5% BT, and PLA 10% BT substrates even though, in the latter two, the contact angle values slightly increased. PLA 15% BT showed an improvement in hydrophobicity almost up to a superhydrophobic surface. In particular, this class of composite showed remarkable variation after 5 and 10 min of aging compared to PLA 10% BT and PLA 5% BT, respectively.

EDX mapping, collected at 500× magnification, permitted to label the distribution of powders through the detection of carbon (red color), as the major components of fibers and barium (blue color), for the BaTiO_3_ filler (Figure 4, Figure 5 and Figure 6, respectively). 

The blue spots were significantly observable, especially in fibers with 10% and 15% of BT powder. In PLA 5% BT, the fillers were concentrated in a few areas and this did not change during the aging test and influenced the morphology and the stability of the fibers. In fact, especially the PLA matrix and PLA 5% BT after 10 min showed several broken fibers. In the other two classes of samples, despite undergoing the same treatment, the same changes did not occur. Furthermore, some fibers were subjected to UV/O_3_ for 10 min attached together, forming some big bundles with the same orientation but presenting less BT on the surface. The diameter and the density of fibers have been calculated and reported in Figure 7.

An increment of fiber diameter is displayed, and this is related to the amount of BT incorporated in the fibers. In fact, PLA matrix samples showed the lowest values compared with the other functionalized samples. Concurrently, the fiber density displayed an opposite trend with a significant reduction of the number of fibers per µm in the samples with a higher amount of filler (PLA 10% BT and PLA 15% BT). Analyzing the diameter and the density separately, no statistical differences emerged inside each class of samples, indicating that the aging test did not affect these parameters.

The plot displayed in Figure 8 shows that PLA 15% BT had the highest potential, while PLA matrix and PLA 5% BT presented the lowest one. The addition of BT dispersed in the matrix conferred a sharp increase of positive potential, especially with 10% and 15% of fillers. The aging test influenced the charge: a reduction of values was visible after 10 min in PLA 5% BT and PLA 15% BT, and after 5 min in PLA 10% BT. No variations were detected in the sample of PLA without BT.

### 2.2. Toxicity in HDF Cells Set on Different Substrates

Figure 9 shows changes in the viability of human dermal fibroblasts (HDF) cells exposed to the different surfaces modified or not by UV/O_3_. The viabilities did not vary consistently between all the samples. Data calculated for all the samples indicated the presence of cells displayed by a slight increase of OD values compared with the negative control.

The PLA 5% BT treated for 10 min under UV/O_3_ presented the highest cell viability, statistically significant compared with most of the other samples (except for PLA 10% BT at 5 and 10 min, and PLA 15% BT treated for 10 min). The effect of the aging test on the cell viability was not detected between the surfaces of each class except for PLA 5% BT.

### 2.3. Bacterial Tests: Microbial Viability Assay and Fluorescence Microscope (CTC-Staining)

As shown in Figure 10, bacterial cell viability CTC was performed at two different times, 12 and 24 h, respectively. At 12 h, PLA 5% BT untreated showed the highest OD value, indicating the largest living bacteria number among the samples; PLA untreated presented a high OD comparing with the rest of the samples. However, bacterial viability decreased drastically in these two types of samples when the surface was subjected to UV/O_3_. The viability also decreased over time, especially for PLA 5% BT, where a statistical difference was observed between the samples after 5 min and then after 10 min. In the case of PLA 15% BT, the trend was opposite with an increase of OD values related to the sample treated for 10 min comparing with the other two samples.

PLA 10% BT samples presented a low amount of bacteria, and no differences emerged between each other. At 24 h of exposure, the highest OD value belonged to PLA untreated, whereas PLA 10% BT and PLA 15% BT displayed the lowest OD values. PLA 15% BT treated for 5 min had a higher amount of bacteria than the untreated and treated for 10 min. However, the values were lower compared with PLA 5% BT and also PLA. PLA 5% BT had intermediate OD values, which did not show any statistical differences in the aging treatment. Another aspect that emerged was a clear reduction of bacterial viability on PLA surfaces subjected to UV/O_3_ at 5 and 10 min. Note that the OD values samples with 10% and 15% of fillers were lower at 24 h than at 12 h. Only PLA presented an increase for all the OD values. In the case of PLA 5% BT samples, the untreated and the one subjected to 5 min of UV/O_3_ showed a drastic decrease, while for the sample treated for 10 min, the OD was higher at 24 h than 12 h.

Fluorescent micrographs were obtained for all the samples after staining them with a CTC dye to evaluate the microbial respiratory activity. The images collected with a magnification of 4× for all the samples (shown in Figure 11 and Figure 12) related to *Staphylococcus epidermidis* exposure for 12 h and 24 h, respectively.

At 12 h, the three PLA matrix samples presented the highest concentration of bacteria. The red spot size appeared bigger on PLA than the other samples. PLA 15% BT surfaces, independently by the aging tests, showed a random distribution of few spots, while in the case of PLA 5% BT and PLA 10% BT the living bacteria were detected in localized areas forming aggregated clusters. At 24 h, the PLA matrix experienced a clear increment of bacterial amount connected with proliferation and biofilm formation. All the PLA surfaces, independently from the aging treatment, were covered by bacterial biofilm. The other samples, at 24 h, showed a higher amount of red-stained areas comparing with the one collected at 12 h, which indicates a slow bacterial proliferation. All the PLA with BT presented localized areas of bacteria growing around the surfaces, but these were not dimensionally comparable with the ones detected on the PLA matrix.

Figure 13 gives a quantitative assessment of the fluorescence microscopy experiments in terms of the percentage of area covered by living bacteria at exposure times of 12 and 24 h for the different tested substrates. At 12 h, PLA matrix samples presented the highest amount of area covered by bacteria. The aging test did not bring any significant changes that could affect the initial bacterial adhesion and proliferation. A statistical reduction was identified on PLA 15% BT treated at 10 min compared with the other composites treated at the same time. The aging test influenced only the PLA 5% BT, particularly the substrates treated for 5 min, which highlighted a reduction of living bacteria, while for ones treated for 10 min, the bacterial amount was in the same range of the samples not subjected to the aging test. Another important aspect regards the statistical reduction of bacteria after 15 min of treatment between PLA 15% BT and the other PLA scaffold incorporating BT.

At 24 h, no statistical differences emerged between the fibers functionalized with BT and treated under UV/O_3_. Between 12 and 24 h, a general increase of bacterial area coverage occurred, indicating cellular proliferation on PLA 5% BT and especially on PLA matrix. In the case of the other two composites, even if a slight no-significant increase occurred, no statistical differences were displayed.

## 3. Discussion

The present work analyzed the morphological and physical changes that occurred with incorporating BT inside PLA fibers. The preparation and chemical and thermal characterization of these samples were reported in a previous study [32]. In vitro tests have been performed to verify the bacteriostatic behavior of the fibers after the aging test. The addition of BT brought different morphological modifications starting from the increase of fiber diameter with the concurrent decrease of fiber density (as shown in Figure 7). The effect of the filler on the morphological properties of fibers has been described in other works, which are in accordance with the results presented a dependence between fiber diameter and amount of filler incorporated [33,34,35,36,37,38]. Apparently, observing the confocal microscope images, an alignment of fibers occurred connected with an increase of BT percentage used. This orientation started to be visible in PLA 10% BT, but it is not clearly observable as in the PLA 15% BT. The reason can be identified in the high viscosity by adding BT particles (as seen in the previous study). This has also been demonstrated by Afifi et al.; their results analyzing different solutions of PLA showed how the increasing of the viscosity led to the formation of homogeneously aligned fibers avoiding the formation of beads [39]. Also, other researchers confirmed how this parameter plays an important effect in the formation of fibers having an ordered pattern [40,41,42]. The fiber alignment, connected with the fiber diameter, could have influenced the roughness making the surface morphologically more homogeneous and less rough, despite the presence of BT. The orientation of the fibers seems to have also influenced the wettability of the samples with the highest percentage of filler incorporated. This change in highly hydrophobic surfaces can be produced by introducing submicron roughness (provided by BT particles) into an inherently hydrophobic surface and by other morphological parameters like orientation. These changes have already been reported in the literature by different researchers [43,44,45,46,47]; the surface potential was also related to the presence of BT. The positive surface potential increased significantly but differently from the other properties; each class of composite presented a variation connected with the aging test. The reduction of potential over time was more evident when increasing the BT amount. PLA with the higher amount of filler presented a remarkable decrease of surface potential after UV/O_3_ exposition, probably due to a modification of the surface. As seen in previous work [32], PLA 10% BT, and especially PLA 15% BT, exhibited a blending of different fibers having the same alignments and forming big clusters. This fusion led to a reduction of BT exposed on the surface and influencing the potential. The aging test that was applied did not significantly influence the features of samples for each class of fibers. Average roughness, fiber diameter, fiber net density, and wettability were not influenced by exposing the samples under UV/O_3_ at different times. All the samples indicated a higher OD value connected with fibroblast adhesion than the negative control, but except for the PLA 5% BT treated for 10 min, no differences were reported, despite the increase of BT. Despite the presence of the filler at different concentrations, the morphology and the wettability did not provide any cellular growth after fibroblasts exposure. In fact, the BT in other works have been indicated as a potential biocompatible material presenting low cytotoxicity and good cellular adhesion [48], but the high hydrophobicity of PLA matrix and composites surfaces represented an obstacle that limited the action of the filler.

The bacterial test results showed a long-term antibacterial effect for samples containing BT at different concentrations against *S. epidermidis*. This is highlighted by WST results (Figure 10), especially in the two classes of composites with higher amounts of BT. The effect of filler was visible on 24 h more than on 12 h. In addition, the CTC images confirmed how the addition of BT provided an improvement in antibacterial properties. The images of the area covered by bacteria showed how the bacterial distribution on PLA 5% BT surfaces was in the same range of the PLA 10% BT and lower than the PLA matrix also at 12 h. The PLA 15% BT displayed the lowest amount of area covered at 12 h, and the values remained unchanged at 24 h. The antibacterial effects of BT were already reported in the literature against both Gram-positive and Gram-negative strains, and it was observed that the bacterial inhibition effect was dependent on the number of particles used [49]. The use of BT as filler in the creation of antibacterial composites has been described by Marin et al., reinforcing scaffolds of polyvinyl-siloxane scaffolds (PVS). Testing the scaffolds against *S. epidermidis*, the main bacteriostatic behavior was associated with the release of Ba^2+^ and the consequent formation of TiO_2,_ which could contribute to the production of hydroxyl radicals (•OH) and free radicals (O^2−^) [50,51]. Furthermore, the slight variation of roughness (especially *Ra*), which occurred due to the addition of BT, could have affected the bacterial adhesion and proliferation. In fact, the accumulation of microorganisms and the development of biofilm depends on the morphological features provided by supports and it is favored especially on rough surfaces. Despite the presence of BT detected by EDX and SEM [1], which could have brought an increase of nanoscale roughness on the surfaces of the fibers, the fillers provided a new fiber rearrangement which decreased the irregularities [52]. The slight variation of wettability did not play a remarkable role in bacterial adhesion.

The surface potential is another important physicochemical property that is involved in the interaction between the samples and the bacteria. In this work, BT generated an increase of positive surface potential, which could have helped to neutralize the bacterial viability. In the other studies, the effect of the surface potential of metal oxide nanoparticles against different bacterial strains has been demonstrated, and it mainly depended upon the interfacial potential, which resulted from the interaction of the particles with bacterial membrane [53]. In the present work, the increase of positive surface potential could have led to a surface neutralization of bacteria triggering the production of ROS, which can cause membrane and DNA damages [54,55]. The surface neutralization occurred because the BT particles took the interfacial potential present between the surface (positively charged) and bacteria (negatively charged) to neutral by the surface functional groups located on the interacting partners [56]. The increase of filler, connected with an increase of positive surface potential favored the generation of these electrostatic forces. From the neutralization, the energy released helped to generate ROS (by Ba^2+^ and TiO_2_) or membrane tension leading to bacterial membrane damages.

## 4. Materials and Methods

### 4.1. Sample Preparation

The procedure to obtain the PLA fibers with the BaTiO_3_ by centrifugal spinning and part of the characterization related to chemical, thermal and mechanical properties (FTIR, SEM, XPS, DSC, TGA) monitoring their variations after the effect of aging tests were explained in our previous work [32]. Briefly, after the production, the samples used for biological tests and characterization were cut in the shape of round disks (0.8 ± 0.06 µm) and sterilized by UV.

### 4.2. Sample Characterization

The water contact angles of the composite fibers were measured using a contact angle machine (Phoenix 300, Kromtek Co., Selangor, Malaysia) at 20 °C and were further evaluated using the Image software (Rasband, W.S., ImageJ, National Institutes of Health, Bethesda, MD, USA). Five microliters of deionized water were dropped onto the surface of the composite fibrous mats. The average water contact angle value was obtained by measuring at least five different places across the same sample surface.

Surface morphology was characterized at a micrometric level with a confocal scanning laser microscope (Laser Microscope 3D and Profile measurements, Keyence, VKx200 Series, Osaka, Japan) capable of high-resolution optical images with depth selectivity. Assembled maps of each sample were collected at 150× magnifications with numerical apertures between 0.30 and 0.95. The instrument was equipped with an automated x-y stage and autofocus function for the z-range. In order to calculate average roughness (*Ra*) and mean roughness depth (*Rz*) values, 25 images randomly acquired from each mapped surface were used.

Scanning electron microscopy (SEM) and Oxford Instruments energy dispersive X-ray spectroscopy (EDX) (SM-700 1F, JEOL, Tokyo, Japan) were used to acquire high-resolution images in order to understand the morphology and to identify the chemical composition of the fibers treated with BT. The substrates were collected on the SEM sample base and then sputter-coated with a layer (15 Å) of platinum before being measured with FE-SEM at an accelerating voltage of 15 kV using different magnification. The diameters of the fibers were obtained from SEM images collected at 1000× for 200 fibers and using image processing software. Fibers density has been calculated by analyzing five images at 1000×. The EDX maps have been collected at 500× magnification and an accelerating voltage of 15 kV. For each class of samples, three different surfaces have been used.

The surface potential of the different fibers was detected by an electrostatic voltmeter operating in a compensation setup. The instrument performed non-contact surface voltage measurements in the range of 0 to ± 3 kV DC or peak AC (Trek, 347). The samples (*N* = 5 for each class), having the same dimensions (1 × 1 cm), were at the same distances from the probe (2 mm).

### 4.3. Fibroblasts and Bacterial Culture

To evaluate the cells’ adhesion, human dermal fibrobalsts (HDFs) were seed on each sample (PLA and PLA/BaTiO_3_) at 10,000 cells and cultured for seven days. All samples were transferred to a new culture plate and added to fresh complete medium.

2-(2-Methoxy-4-nitrophenyl)-3-(4-nitrophenyl)-5-(2,4-disulfophenyl)-2H-tetrazolium monosodium salt (WST-8) solution (Cell Count Reagent SF, Nacalai Tesque, Kyoto, Japan) was added to the culture at a final concentration of 10%. After one hour of incubation at 37 °C, the supernatant was transferred to a 96-well plate, and the absorbance of each well was measured with a microplate reader (Emax; Molecular Devices, San Jose, CA, USA) at 450 nm against 650 nm as a reference. For each class, four samples have been used and compared with negative control (-) presenting culture medium without cells.

Freeze-dried pellets of Gram-positive *Staphylococcus epidermidis* (ATCCTM 14990^®^) (*S. epidermidis*, henceforth) were hydrated in heart infusion (HI) broth (Nissui, Tokyo, Japan) and incubated at 37 °C for 18 h in brain heart infusion (BHI) agar (Nissui). The mixture was subsequently assayed for colony forming units (CFU) and diluted to a concentration of 1 × 10^8^ CFU mL^−1^ using phosphate-buffered saline (PBS). The bacterial suspension was then transferred in 100 μL aliquots onto Petri dishes containing the substrate samples embedded in BHI medium and incubated at 37 °C under aerobic conditions for 12 and 24 h. Then, cell viability was evaluated by a tetrazolium-based assay using the Microbial Viability Assay Kit (WST-8; Dojindo Laboratories, Kumamoto, Japan). Substrates were collected and soaked in 1000 μL of PBS in 12-well plates. WST-8 solution was added to each well and OD values measured (the absorbance at 490 nm) using plate reader EMax (molecular devices, San Jose, CA, USA) after incubation for 30–60 min.

At each time point, three samples for each class were observed by fluorescence microscopy (BZ-X700; Keyence, Osaka, Japan). Bacteria were stained with 5-cyano-2,3-ditolyl-2 H-tetrazolium chloride (Bacstain-CTC rapid staining kit, Dojindo laboratories, Kumamoto, Japan) to detect bacterial respiration. After incubating at 37 °C for 30 min, the disks were fixed in 4%-Paraformaldehyde Phosphate Buffer solution (Nacalai Tesque). Subsequently, the samples were analyzed by a fluorescence microscope using 4× of magnification. All the images collected have been subsequently processed and the areas associated with bacterial viability have been counted using Image software (Rasband, W.S., ImageJ, National Institutes of Health, Bethesda, MD, USA).

### 4.4. Statistical Analysis

The experimental data were analyzed with respect to their statistical meaning by computing their mean value ± one standard deviation and using and two-way ANOVA with Tukey’s post hoc analysis for pre-test characterization (morphology, contact angle, and electrical analysis) and for biological assays (microbial viability assays, cytotoxicity and fluorescence images); *p* < 0.05 was considered statistically significant and labeled with an asterisk.

## 5. Conclusions

The study of antibacterial properties of a composite made by PLA fibers filled with different concentrations of BT have been provided. The samples were subjected to UV/O_3_ and subsequently tested with *S. epidermidis* to verify the fillers’ effect and possible modifications that could occur after the aging tests. The data collected before and after the in vitro tests brought out the following conclusions:

The addition of BT influenced the morphological properties of the fibers. Increasing fiber diameter with a concurrent decrease of density of fiber nets, as a reduction of roughness associated with the development of a more oriented distribution, occurred with the presence of BT. The particles were homogeneously distributed on the fibers also after the aging test.

The incorporation of BT influenced the physical properties bringing an increase of positive surface potential, but consequently, did not make any significant variation regarding the wettability of the surfaces that presented a high hydrophobicity.

Tests in vitro indicated how the BT influenced the bacterial viability over time, preventing the bacterial adhesion and proliferation. Furthermore, no differences were detected in analyzing the cytotoxicity of the substrates.

The stability associated with long-term antibacterial effects suggests the possibility of using this composite for the production of protective textiles, and wound dressing applications.

## Figures and Tables

**Figure 1 molecules-26-02918-f001:**
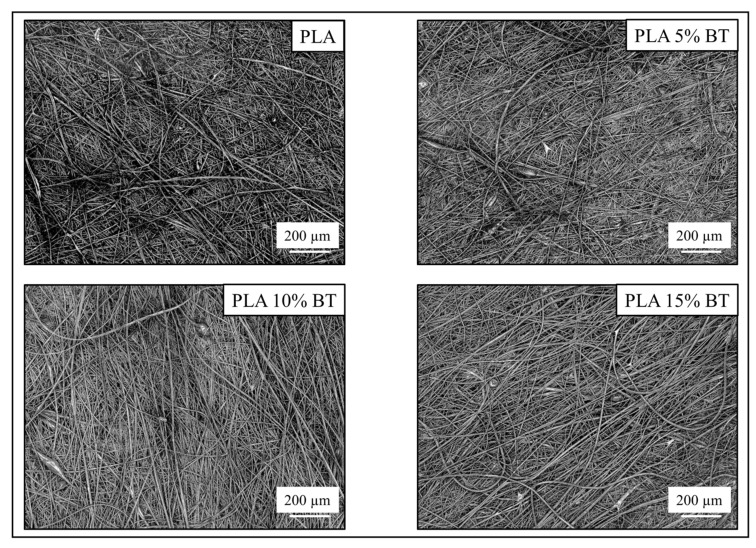
3D laser micrographs of the four different substrates obtained before the aging treatment at 150× of magnification.

**Figure 2 molecules-26-02918-f002:**
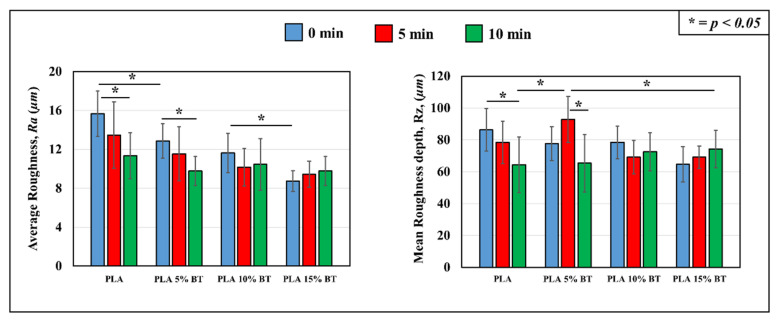
Comparison of roughness parameters, Ra and Rz of the substrates samples before and after treatment. The values are obtained by calculating the parameter in 25 images randomly acquired from each mapped surface using 150× of magnification.

**Figure 3 molecules-26-02918-f003:**
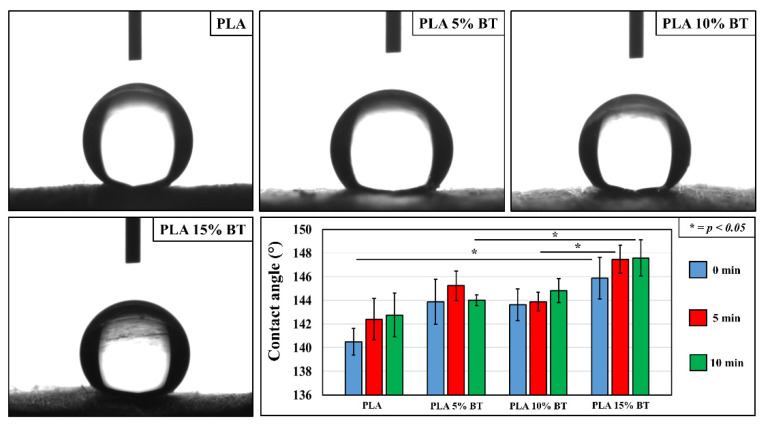
Images of water contact angle measurements on Polylactic acid (PLA) fibers and composites before the aging test. The graph represents the value of the contact angle for all the substrates analyzed before and after the UV/O_3_ exposure. For each substrate, five different points have been tested.

**Figure 4 molecules-26-02918-f004:**
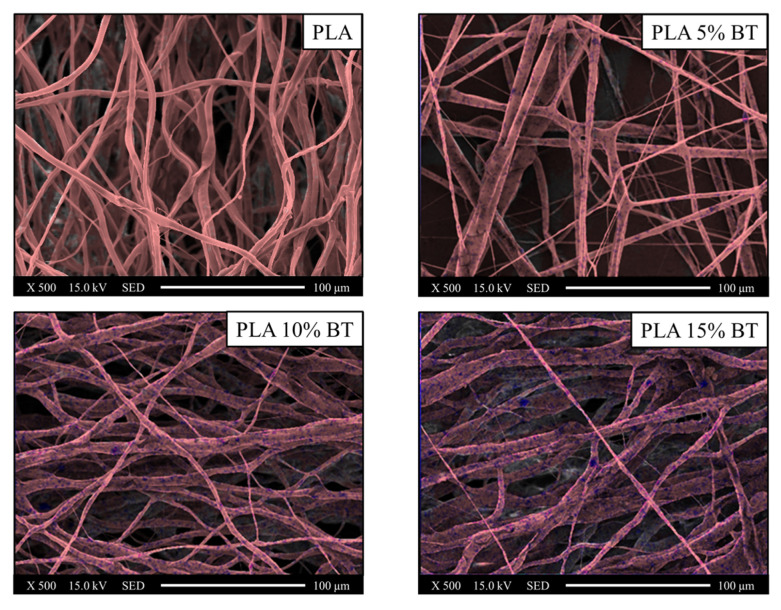
Images collected using EDX elemental maps of the PLA fibers and composites before aging tests and converted in two different colors (red for carbon and blue for barium) and overlaid on the SEM images in order to verify the incorporation of barium titanate inside the fibers and the distribution. EDX and SEM images were collected at 500× magnification.

**Figure 5 molecules-26-02918-f005:**
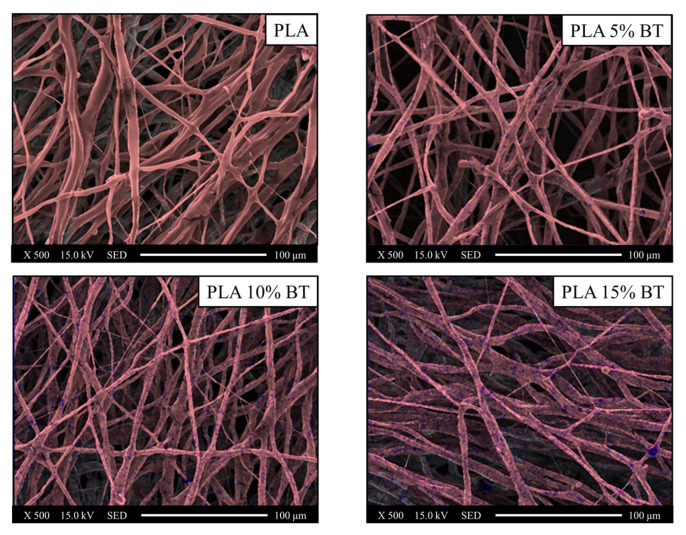
Images collected using EDX elemental maps of the PLA fibers and composites after five minutes of UV/O_3_ exposure and converted in two different colors (red for carbon and blue for barium) and overlaid on the SEM images in order to verify the incorporation of barium titanate inside the fibers and the distribution. EDX and SEM images were collected at 500× magnification.

**Figure 6 molecules-26-02918-f006:**
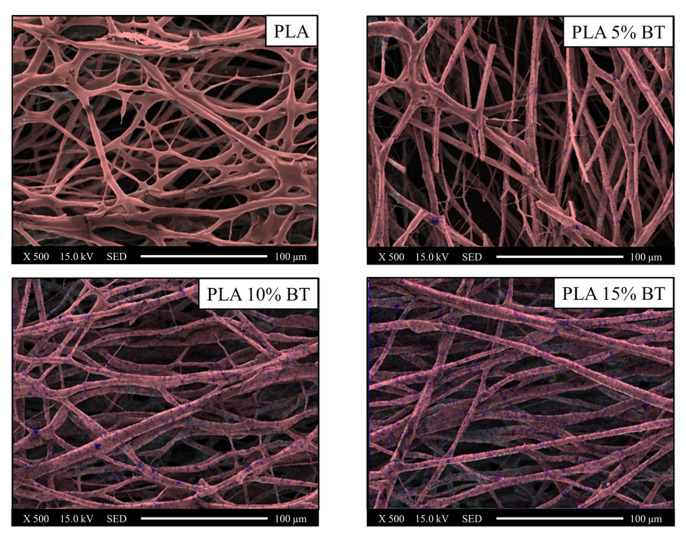
Images collected using EDX elemental maps of the PLA fibers and composites after ten minutes of UV/O_3_ exposure and converted in two different colors (red for carbon and blue for barium) and overlaid on the SEM images in order to verify the incorporation of barium titanate inside the fibers and the distribution. EDX and SEM images were collected at 500× magnification.

**Figure 7 molecules-26-02918-f007:**
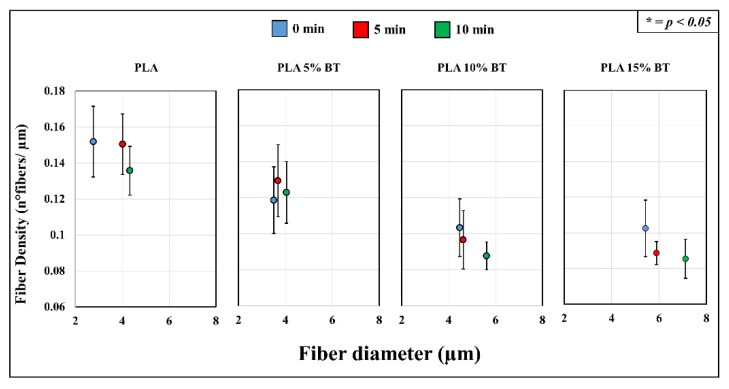
Fiber density and diameter of the PLA matrix and composites calculated using five images for each class of substrates collected by SEM at 500× magnification.

**Figure 8 molecules-26-02918-f008:**
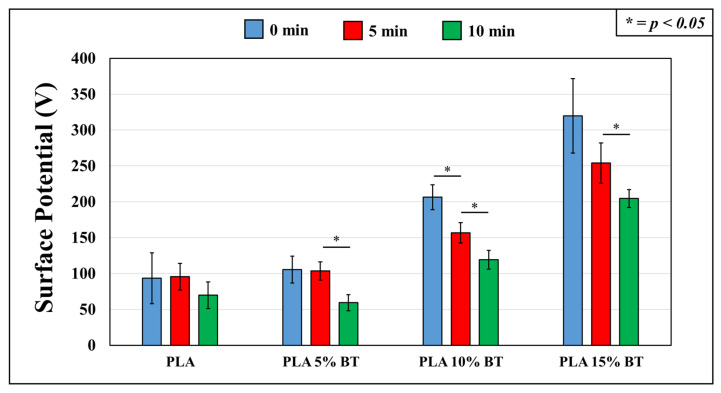
The surface potential of the substrates obtained before and after aging treatment. For each substrate, five samples have been used and maintained at a distance from the probe (2 mm).

**Figure 9 molecules-26-02918-f009:**
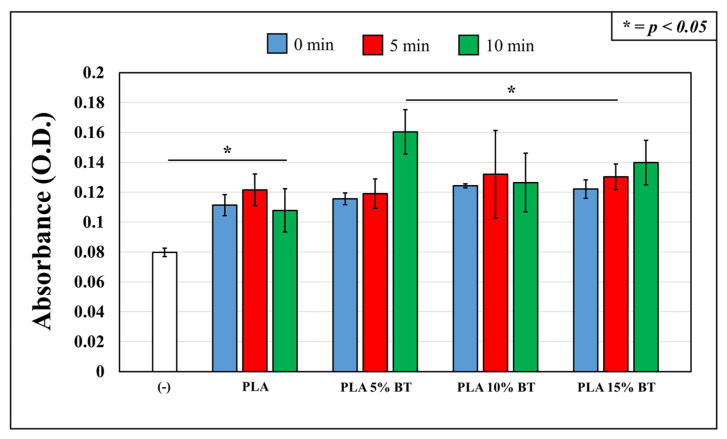
Cell viability assessed by Cell Count Reagent SF kit on the four samples before and after aging treatment using human dermal fibroblasts (HDF) cells and validated by statistical analysis.

**Figure 10 molecules-26-02918-f010:**
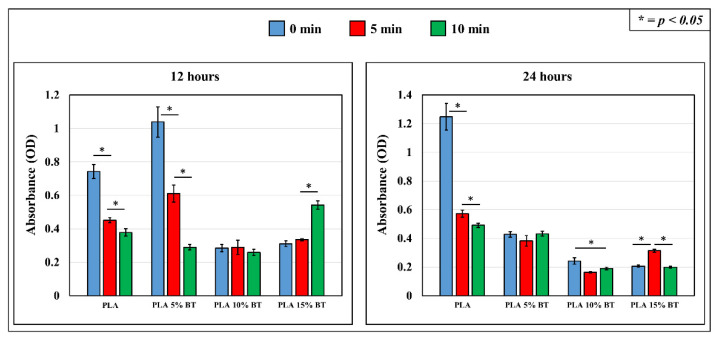
Experimental time-lapse results and statistical validation of optical density (O.D.) values related to the microbial viability test performed on the four samples after 12 and 24 h *Staphylococcus epidermidis* exposure.

**Figure 11 molecules-26-02918-f011:**
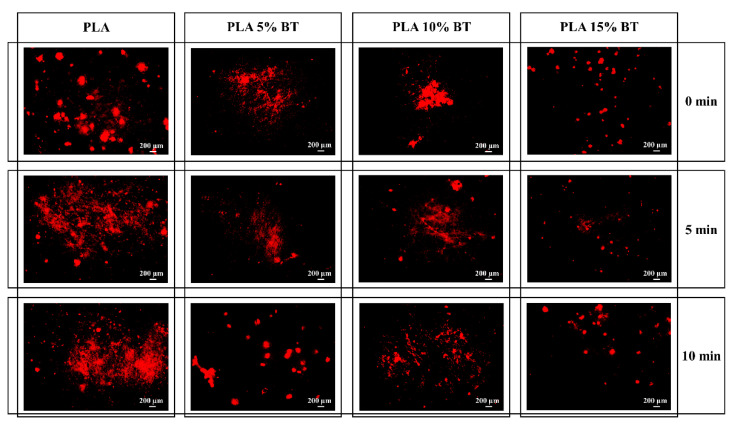
Fluorescence micrographs after CTC staining of *S. epidermidis* exposed for 12 h to the four substrates. Images were obtained at a 4× magnification.

**Figure 12 molecules-26-02918-f012:**
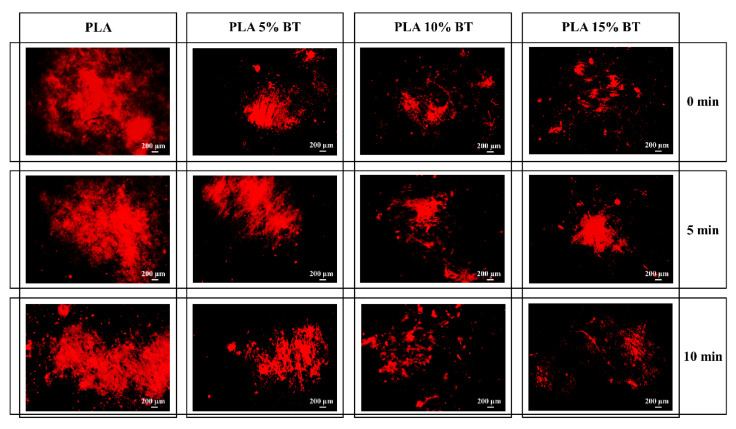
Fluorescence micrographs after CTC staining of *S. epidermidis* exposed for 24 h to the four substrates. Images were obtained at a 4× magnification.

**Figure 13 molecules-26-02918-f013:**
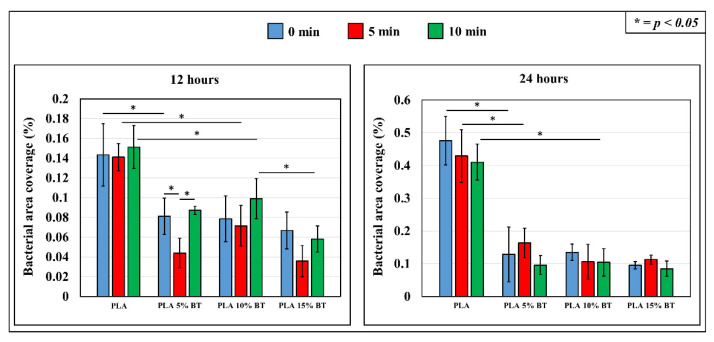
Quantitative assessment of fluorescence microscope images of the four samples collected at *S. epidermidis* exposure time of 12 and 24 h and validated by statistical analysis.

## Data Availability

The data presented in this study are available on request from the corresponding author.

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
