# Peer review of "Bacteriostatic Behavior of PLA-BaTiO_3_ Composite Fibers Synthesized by Centrifugal Spinning and Subjected to Aging Test"

_molecules, 2021, doi:10.3390/molecules26102918_

Round 1
Reviewer 1 Report
The paper presents experimental results of the antibacterial properties of PLA fibres with BT fillers, and under UV/O3 aging tests. The novel aspect of the work is the study of the antibacterial properties of the matrix with different levels of BT filler added, and these data are well presented.
1. there is an issue with the ordering in this paper – Results is Section 2 but begins with 3.1, 3.2, 3.3, then Section 3 is Discussion;
2. the methods are at the end in Section 4 which is unusual, there is little value in discussing results prior to explaining how the experiments were prepared and undertaken. I suggest moving Methods after the Introduction, as is standard practice, and correcting the Section numbering
3. much of the paper describes the PLA + BT structure, and the effect of UV ageing, prior to describing the bacteriostatic characteristics – however, the authors refer to their other paper [28] as also describing these structural characteristics. Ref [28] is referenced as “Under Review”, therefore this reviewer could not ascertain what/if any overlap exists between these papers - perhaps the authors could clarify this. This is also notable in the conclusions, where only one conclusion actually relates to the bacterial study.
4. there is much discussion of statistical results however the actual statistics are not detailed, while these are expected to be inferred from the asterix and lines shown in the Figures – please fully explain these asterix and lines nomenclature so that the statistical results can be understood clearly by the reader
5. there are several instances in the discussion where vague/qualitative terms are used such as “drastically” (line 277), “huge decrease” (line 280), “clearly visible” (line 298), etc. Please replace these with quantitative comparisons
Author Response
We sincerely thank the Reviewer for his/her patience and the continuous improvement provided by his/her suggestions. Our reply and corrections are shown below. We believe to have clearly responded to all the Reviewer’s queries

Reviewer 2 Report
This manuscript reports the characterization of PLA-BaTiO3 composite fibers synthesized by centrifugal spinning. The paper addresses an important current problem regarding preparing effective antibacterial materials in medical field. The manuscript is well drafted and scientifically sound. The evaluations of results are sufficient, clear and contain sufficient information. The Figures are also clear and demonstrative.
My main concerns about this paper as follows:
The Authors should highlight the novelty of this research work and also how these BaTiO3 particles loaded PLA fibers can be used in practice.
The reference list should be supplemented with some more recent papers on this field.
There are many mistypes in the manuscript that should be corrected.
Author Response

(The authors gave the same response as above.)
